# CLARIFY WHEN NECESSARY: RESOLVING AMBIGUITY WITH LANGUAGE MODELS

## ABSTRACT

Resolving ambiguities through interaction is a hallmark of natural language, and modeling this behavior is a core challenge in crafting AI assistants. In this work, we study such behavior in LMs by proposing a task-agnostic framework for resolving ambiguity by asking users clarifying questions. Our framework breaks down this objective into three subtasks: (1) determining *when* clarification is needed, (2) determining *what* clarifying question to ask, and (3) responding accurately with the new information gathered through clarification. We evaluate systems across three NLP applications: question answering, machine translation and natural language inference. For the first subtask, we present a novel uncertainty estimation approach, INTENT-SIM, that determines the utility of querying for clarification by estimating the entropy over user intents. Our method consistently outperforms existing uncertainty estimation approaches at identifying predictions that will benefit from clarification. When only allowed to ask for clarification on 10% of examples, our system is able to double the performance gains over randomly selecting examples to clarify. Furthermore, we find that INTENT-SIM is robust, demonstrating improvements across a wide range of NLP tasks and LMs. Together, our work lays foundation for studying clarifying interactions with LMs.

## 1 INTRODUCTION

Ambiguity is embedded throughout natural language, and even simple utterances can have multiple interpretations when read in isolation. Ambiguity serves a key, communicative function in language, allowing speakers to omit details by relying on information that is inferable from the extra-linguistic context of the conversation (e.g., temporal, social, and physical) (Piantadosi et al., 2012). At times, however, the speaker's intent is still unclear despite the context. In such cases, further interaction is required to resolve the ambiguity, often by asking and answering clarifying questions.

With the recent progress in large language model (LLM) development, interactive AI assistants (e.g., ChatGPT, Claude, LLaMA-2) have risen to prominence in our daily lives; yet, these systems often fail to interact with users to resolve ambiguity in their requests. We address these shortcomings by establishing a task-agnostic framework for modeling and resolving ambiguity with LLMs using clarifying questions. We find that imbuing LLMs with the ability to ask clarifying questions can improve performance on a variety of NLP tasks.

Our framework breaks down the objective of resolving ambiguity into three sequential subtasks, which we depict in Figure 1. In our first task, systems must decide *when* to ask the user for clarification. We evaluate system for this task on their ability to maximize end-task performance while minimizing interaction cost. In our second task, systems must then decide *what* to ask the users. Here, systems should ask questions that expose the ambiguity in the user's requests, eliciting a disambiguating response. Finally, after asking the user a clarifying question and receiving their response, systems perform the third and final task: producing the appropriate output given the ambiguous input and the user's clarification.

We apply this framework to a three of NLP settings: question answering (QA), machine translation (MT), and natural language inference (NLI). To cover each of these applications, we draw examples from existing datasets focused on modeling ambiguity (Min et al., 2020; Bawden et al., 2018; Liu et al., 2023) and use multiple annotations to derive samples from the natural distribution over user intents for each ambiguous input. Having access to these natural distributions enables realistic

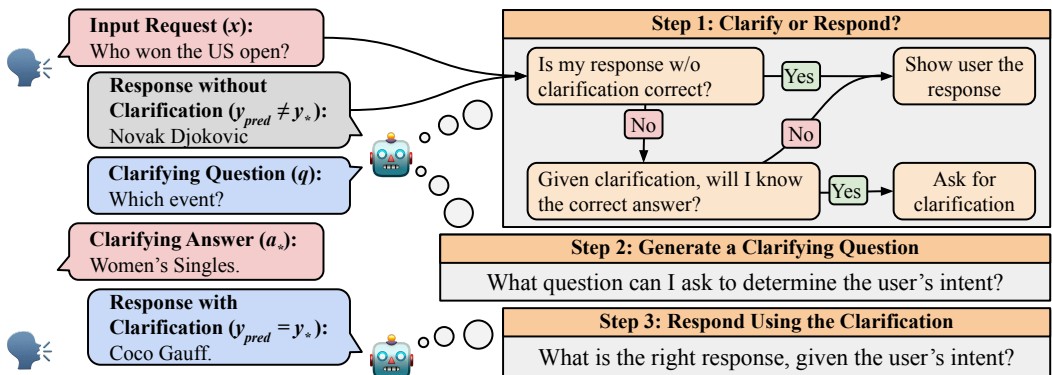

Figure 1: Our three-stage framework for resolving ambiguity with clarification questions. In the first step, systems must identify which inputs will benefit from clarification. In the second step, after deciding to clarify, we provide systems with a clarifying QA pair corresponding to the gold interpretation, which we generate from existing sources of disambiguated input/output pairs. Finally, in the third step, systems use the input and the clarifying QA pair to arrive at the correct output.

evaluations, particularly for determining *when* clarification is needed. Many ambiguous examples, while possessing multiple feasible interpretations, have only one mostly-likely interpretation that dominates the distribution over user intents (e.g., "She's from Boston" typically does not mean "Boston, Georgia"). Systems for our first subtask should, therefore, be evaluated for their ability to identify such cases and avoid asking for unnecessary clarification.

We also develop systems for each of our subtasks, including an oracle method for generating clarification questions along with answers for different intents. We use this oracle to evaluate our other two subtasks, as performance on these tasks depends heavily on the quality of clarifying interactions.

Finally, we conclude our work by introducing INTENT-SIM: a novel method for uncertainty estimation that we use to determine when to ask for clarification. INTENT-SIM involves estimating the entropy over user intents by simulating multiple user-assistant interactions. Through our experiments, we demonstrate the INTENT-SIM consistently outperforms other uncertainty estimation baselines at identifying predictions that are both incorrect and can be improved with clarification. We also find that these improvements are robust across different tasks and LLM systems. When limiting systems to only clarifying 10% of inputs, INTENT-SIM achives the best performance across all 7 LLM-plus-task setting we experiment with in this work.

## 2 A FRAMEWORK FOR RESOLVING AMBIGUITY THROUGH INTERACTION

We begin this work by formally defining three subtasks for resolving ambiguity with clarifying questions: (1) determining *when* to ask for clarification, (2) identifying *what* clarifying question to ask, and (3) reacting to clarification with the proper response. In Figure 1, we depict how each of these three subtasks are sequentially applied in a user-assistant interaction to resolve ambiguity. This figure also depicts the notation used in this work, which we define below.

**Definitions** Each interaction begins with the user providing an initial input request, $x$, to the LLM assistant. Some inputs may be ambiguous, resulting in many feasible output responses for the system to choose from, which we denote as the set $Y = \{y_i\}_1^k$. One of these outputs, $y_* \in Y$, represents the gold output corresponding to the user's intent behind their ambiguous request. To determine the users intent, systems may ask the user a clarifying question, $q$. The user then responds with the clarifying answer corresponding to their intent, $a_* \in A = \{a_i\}_1^k$. For simplicity, we assume a bipartite matching between the sets of clarifying answers, $A$, and feasible final responses, $Y$.

Each input request $x$ has its own distribution over intended interpretations, $\mathbb{P}(y = y_*|x)$. Accurately modeling this distribution is essential for avoiding asking unnecessary clarifying questions. For example, when this distribution is dominated by a single feasible output, systems may want to forego clarification and respond to the user directly. Gathering annotations for the true distribution over intents, however, is intractable and temperamental (i.e., subject to changes over time, location, and

individual preferences). Instead of assuming that we have access to this gold distribution, we say that our dataset consists of $(x, y_*)$ tuples, where intents and their respective outputs, $y_*$, are sampled from this distribution. We describe the data generation process for creating these samples in Section 3.

## 2.1 TASK 1: DETERMINING WHEN CLARIFICATION IS NECESSARY

The frequency with which systems should ask for clarification depends on the demands of the domain and preferences of the user. In high-stake settings, we may want systems to frequently ask for clarification. Likewise, for time-sensitive issues, we may want to minimize the number of interactions. As such, we do not treat determining when to ask for clarification as a classification task; instead, we evaluate this challenge as an uncertainty estimation objective. While standard uncertainty quantification only cares about estimating the performance of a given prediction, our task requires estimating how much performance would increase if provided clarification on the input.

This task requires systems to disentangle the two factors that contribute to model uncertainty: epistemic and aleatoric uncertainty (Cole et al., 2023). Epistemic uncertainty refers to uncertainty that is due to a lack of knowledge. In the tasks we consider, this may occur in questions about an entities the LLM hasn't seen or words it hasn't observed the translation of. Aleatoric uncertainty, on the other hand, refers to uncertainty that is the result of some intrinsic randomness in the output. This randomness is often due to ambiguity, which we resolve through interaction. Systems for this task must identify instances with high aleatoric uncertainty, where the user's intent is ambiguous, and low epistemic uncertainty, where has the knowledge required to respond after clarification.

Concretely, systems for this task must predict a scalar uncertainty estimate, $u(x)$, for each input, $x$, that correlates with how much performance is expected to improve after clarification. Whether predictions improve with clarification is dependent on the performance on the other two subtasks (i.e. the quality of the clarifying interaction and the system's ability to use it to produce the correct output). We address these dependencies in our descriptions of the other two subtasks below.

**Evaluation Metric: Performance Under a Fixed Interaction Budget**   To evaluate this task, we provide systems with an interaction budget, $b \in [0, 100]$, and allow systems to ask clarification questions on $b\%$ of input examples. We use each system's uncertainty estimates, $u(x)$, to determine top $b\%$ of candidate examples to provide clarification for, then evaluate system performance under this interaction budget. This metric is closely related to those used in selective prediction (El-Yaniv & Wiener, 2010), a uncertainty estimation task where low-confidence predictions are either withheld or passed onto a human oracle to annotate by hand (Tran et al., 2022).

**Evaluation Metric: AUROC**   Area under the receiver operating characteristic (AUROC) is a metric that commonly used in standard uncertainty quantification, where it is used to evaluate an uncertainty estimator's ability to classifying correct and incorrect predictions over all possible confidence thresholds. In our setting, we adapt this metric to evaluate the uncertainty estimate's ability to identify whether or not performance on an example will improve with clarification.

## 2.2 TASK 2: GENERATING CLARIFYING QUESTIONS AND ANSWERS

After determining whether or not clarification is needed, the next step is generating a clarifying question to ask the user and receiving their response. Numerous prior works have explored the task of generating a clarifying question based on an input, particularly in classification (Yu et al., 2019), FAQ (Rao & Daumé III, 2018), and moral assesment (Pyatkin et al., 2022) domains. Given the depth of prior work on the subject, we do *not* propose or evaluate new methods for generating clarifying questions conditioned on the input. Instead, we develop an oracle prompting method for generating clarifying questions and answers for different user intents, which we describe Section 4.

The purpose of developing this oracle is to establish a stable test-bed for evaluating systems on the other two subtasks. As mentioned above, the performance of each step in our clarifying pipeline is inextricably linked to the performance of the other two steps: accurately responding to clarifications requires high quality clarifying interactions, and determining when to ask for clarification also depends on the utility of the clarifying interactions. While prior work has established methods evaluating clarifying question generation, our other two tasks are more novel and less well studied in LLMs. Therefore, our focus is on evaluating performance on our first and third tasks in isolation, using our oracle-generated clarifying interactions to limit the dependence on this intermediate step.

## 2.3 TASK 3: RESPONDING TO CLARIFICATIONS

In this final task, systems must use the input and the clarifying question and answer to arrive at the appropriate response. To evaluate this task, we simply evaluate the LLMs generated output $\hat{y}$, which is conditioned on the ambiguous input and the clarifying QA pair, against the gold output $y_*$. We use different metrics for comparing $\hat{y}$ against $y_*$ for each target task, which we describe in Section 3.

We evaluate systems under two data generation processes for sampling $(x, y_*, q, a_*)$ examples. The first, SAMPLED, is our standard setting, using samples from the true distribution of intended interpretations as described above. While this setup is well suited for estimating system performance in realistic settings, it can also underestimate the importance of achieving high performance at the tails of the distributions over intents. To avoid over-indexing on only the most common interpretations, which may lead to misleading and biased responses as a whole, we introduce the second setting, UNIFORM, where we evaluate on all interpretations of each input, weighing each equally.

## 3 DATASETS AND APPLICATIONS

We apply our framework to three tasks and datasets for modeling ambiguity. All datasets label ambiguous inputs with their different interpretations, given as disambiguated rewrites or as different contexts, along with their respective outputs. We use these annotations later in developing our oracle system for generating clarifying questions and answers. All datasets lack existing labels for the distribution over these intents. Below, we describe each dataset in detail, as well as our methods for sampling intents for each example. We include dataset details in Appendix A.

### 3.1 QUESTION ANSWERING

We use the AmbigQA (Min et al., 2020) dataset, which re-annotates questions from NaturalQuestions (Kwiatkowski et al., 2019) with whether they are ambiguous. For each ambiguous example, they also annotate different intents as disambiguated revisions of the initial question, paired their respective answers. To draw from the true distribution over intents, we use the original annotated answers from NaturalQuestions as samples of intended outputs, $y_*$. We then map these sampled outputs to their respective intents by identifying which disambiguation contains the same answer.

**QA Performance Metric** We evaluate performance for QA using answer recall, measuring whether the gold answer string appears in the LLM's generated output after normalization (Chen et al., 2017). This deviates slightly from prior work Rajpurkar et al. (2016) that evaluates for strict exact match after normalization, as chat-based LLMs to generate verbose, sentence-length outputs as opposed to short answers (e.g., "The stern is the back of the boat." instead of "the back").

### 3.2 NATURAL LANGUAGE INFERENCE

We source NLI data from the AmbiEnt dataset (Liu et al., 2023), which consists of ambiguous premise/hypothesis pairs that are paired with disambiguated revisions for each of their feasible interpretations. Annotators for this dataset are first presented with the ambiguous input and are asked to label it as an NLI example. Annotators are then shown the different disambiguations each input, and are asked to label each interpretation again. We use these multiple annotations to identify which interpretation's label is consistent with the label annotators gave the initial, ambiguous input. We then use the matching interpretation as our sampled user intent and output label, $y_*$.

**NLI Performance Metric** We evaluate systems using standard 3-way (entailment, contradiction, neutral) classification accuracy.

### 3.3 MACHINE TRANSLATION

The meaning of a sentence can be ambiguous when presented in isolation, but becomes clear in its document-level context. Rich previous works have explored when sentence-level translation to fail in context (Lopes et al., 2020; Yin et al., 2021; Voita et al., 2019). We source examples of ambiguous translations from the DiscourseMT dataset (Bawden et al., 2018), a manually crafted test set of ambiguous English-French translations. Each example consists of an ambiguous test sentence

Table 1: Example instances for each task for different ambiguity types, along with what proportion of ambiguities in each dataset fall into each type from our manual analysis on 150 examples.

| Task | Ambiguity Type | Input ($x$) and Clarifying Question ($q$) | Proportion |
|------|----------------|-------------------------------------------|------------|
| QA | Word-Sense Disambiguation / Entity Linking | $x$: Who wins at the end of friday night lights? $q$: Are you referring to the Friday Night Lights film, book, or television series? | 48% |
| | Literal vs. Implied Interpretation | $x$: Real name of gwen stacy in amazing spiderman? $q$: Are you asking for the name of the actress who plays Gwen Stacy, or the full name of the character Gwen Stacy? | 8% |
| | Multiple Valid Outputs | $x$: When did west germany win the world cup? $q$: Which time? | 44% |
| NLI | Word-Sense Disambiguation | $x$: Every night, the baby is fed milk. / Some nights, the baby is fed milk. $q$: Does the baby get fed milk every night or just some nights? | 44% |
| | Literal vs. Implied Interpretation | $x$: The cake was so dry, it was like eating sand. / The cake was so dry, it was inedible. $q$: Was the cake not suitable for eating or not safe to eat? | 56% |
| MT | Word-Sense Disambiguation | $x$: It's a little steeper than I was expecting. $q$: What kind of mole are you referring to? | 100% |

paired with two possible context sentences, where the translation of the test sentence depends on which context sentence precedes it. We use these test sentences, without context, as examples of ambiguous user inputs, taking its two possible translations as the set of feasible outputs. We also include the context sentences, which are only annotated with one feasible translation each, as examples of unambiguous inputs. While this dataset does not contain annotations for estimating distribution over interpretations, sentences in this dataset are hand-crafted to be highly ambiguous. We, therefore, simply use the uniform distribution over interpretations in our experiments.

**MT Performance Metric** We evaluate using contrastive accuracy (Maruf et al., 2019). This binary metric measures whether an LLM assigns a greater likelihood to the intended translation of an ambiguous sentence over the alternative. For unambiguous examples, we simply say that the system gets the interpretation correct without clarification. We deviate from the standard MT metrics (e.g., BLEU), as confounding factors such as variance in sentence structure often overshadow the word-level, semantic differences between translations.

## 4    AN ORACLE FOR GENERATING CLARIFYING QUESTIONS

We begin our discussion of systems, experiments, and results by introducing our oracle method for generating clarifying questions, which we use to establish a test bed for evaluating system on the other two tasks our pipeline. Our oracle makes use of few-shot prompting with GPT-3.5 (OpenAI, 2022), providing systems with instructions and two hand-written exemplars to accomplish the following task: Given the ambiguous input, $x$, and its different interpretations, each corresponding to a different output $y \in \{y_i\}_1^k$, systems must generate (1) clarifying question differentiating each interpretation, $q$, and (2) then a clarifying response, $\{a_i\}_1^k$, corresponding to each interpretation. The format of the different interpretations used as input to this system depend on the available annotations in each dataset: we use disambiguated revisions of $x$ for QA and NLI and the different target translations, $\{y_i\}_1^k$, for MT. This is an oracle setting, as it requires access to the different feasible interpretations of each input. We minimally edit the prompts between each tasks to reflect the different inputs and and interpretation formats from each task. See Appendix B for prompts and details.

**Clarifying Interaction Analysis** In Table 1, we identify the most common causes of ambiguity by analyzing clarifying questions. The most common cause across all tasks is word-sense disambiguation. In QA, where named entities are more common, this also commonly surfaces as entity linking ambiguities. The second cause is due to the literal and implied interpretations of each input. In QA, this usually occurs when a question literally means something different from what the user probably meant to ask. In NLI, we find that this frequently occurs due to figurative language, where it is un-

Table 2: Performance on responsiveness to clarification. We evaluate three settings: providing the clarifying QA pair (Follow), disambiguated input (Disambig), and baseline (Direct) without clarifying information. For QA and NLI, we evaluate under two different data generation processes, either uniformly weighing all interpretations or using our sampled interpretations. We evaluate MT using contrastive accuracy, QA using EM accuracy, and NLI using 3-way classification accuracy.

| Model | Clarification | MT | QA | | NLI | |
|---|---|---|---|---|---|---|
| | | Uniform | Uniform | Sampled | Uniform | Sampled |
| GPT3 | Direct | 50.0 | 22.7 | 51.8 | 31.2 | 41.7 |
| | Follow | 85.8 (35.8) | 40.8 (18.1) | 61.8 (10.0) | 31.6 (0.4) | 45.9 (4.2) |
| | Disambig | 84.7 (34.7) | 41.2 (18.5) | 62.0 (10.2) | 30.6 (-0.6) | 30.6 (-11.1) |
| LLAMA2 7B | Direct | 50.0 | 14.5 | 31.4 | 29.4 | 32.4 |
| | Follow | 46.6 (-3.4) | 27.3 (12.8) | 45.4 (14.0) | 25.4 (-4) | 35.9 (3.5) |
| | Disambig | 45.5 (-4.5) | 25.7 (11.2) | 41.1 (9.7) | 29.8 (0.4) | 29.8 (-2.6) |
| LLAMA2 7B Chat | Direct | 50.0 | 18.1 | 37.3 | 41.0 | 43.5 |
| | Follow | 43.2 (-6.8) | 32.0 (13.9) | 47.9 (10.6) | 55.3 (14.3) | 52.5 (9.0) |
| | Disambig | 44.9 (-5.1) | 26.5 (8.4) | 42.0 (4.7) | 40.0 (-1.0) | 40.0 (-3.5) |
| LLAMA2 13B | Direct | 50.0 | 17.7 | 39.1 | 30.6 | 37.4 |
| | Follow | 46.6 (-3.4) | 34.1 (16.4) | 53.7 (14.6) | 34.6 (4.0) | 43.1 (5.7) |
| | Disambig | 47.2 (-2.8) | 32.4 (14.7) | 50.8 (11.7) | 30.2 (-0.4) | 30.2 (-7.2) |
| LLAMA2 13B Chat | Direct | 50.0 | 17.9 | 40.0 | 28.0 | 40.7 |
| | Follow | 40.9 (-9.1) | 33.5 (15.6) | 50.9 (10.9) | 49.1 (21.1) | 52.5 (11.8) |
| | Disambig | 42.6 (-7.4) | 28.5 (10.6) | 45.2 (5.2) | 26.6 (-1.4) | 26.6 (-14.1) |

clear whether the sentence should be interpreted literally. In MT, however, we find these ambiguities in the source sentence can usually be captured in its translation. The last common cause we find is ambiguity due to multiple valid outputs. This cause only applies to QA where only reporting one answer may mislead users. We do not find this type of ambiguity in MT, where multiple translations of any sentence is a given, nor in NLI, where classes are designed to be mutually exclusive.

# 5 EXPERIMENTS: RESPONSIVENESS TO CLARIFICATION

**Setting** We evaluate LLM variants for their responsiveness to clarifications by comparing their performance on ambiguous examples with and without clarification. We use standard few-shot prompting for all systems (LLaMA-2, LLaMA-2-Chat, GPT-3), providing them with demonstrations from each task with and without the clarifying QA pairs. We use four randomly sampled exemplars for each example and perform greedy decoding. The exact prompts are available in Appendix B.

**Results** We report our results in Table 2. We find that, across tasks and systems, LLMs can leverage clarifying questions and answers to improve their response. One exception to this trend, however, is the performance of LLaMA-2 variants on MT. We attribute this poor performance after clarification to LLaMA-2's low translation performance due to insufficient multilingual pre-training.

Another notable trend is that systems tend to perform better with clarifying questions and answers than with disambiguated inputs, particularly for QA and NLI. We attribute this the way our QA and NLI datasets construct disambiguated interpretations. These datasets create disambiguated revisions of each ambiguous input by applying a minimal set of token-level edits to the initial input. While this makes disambiguations easier to annotate and compare, it comes at the cost of naturalness of the resulting disambiguated sentences. In contrast, our clarifying interactions do not have the same minimal-edit constraints and more closely resemble the pretraining distributions of these systems.

We also find that there is no consistent improvement in LLaMA-2 and LLaMA-2-Chat's ability to use clarifying interactions. While task performance changes as a whole with chat-finetuning, the gains from providing clarifications remains consistent between equal size LLaMA-2 systems. These findings reinforce our motivations for studying this problem, as existing systems struggle to use clarifying questions and existing methods for chat-finetuning do not adequately train this ability.

Table 3: Generations from our INTENT-SIM method. Systems greedily generate a clarifying question based on the input, then sample multiple user responses. We group equivalent responses using an NLI system, then compute the likelihoods and entropy over the grouped, simulated intents.

| Input with Sampled Clarification Question | Simulated User Answers | Likelihood |
|---|---|---|
| $x_{MT}$: There, on the trunk. 

 $q_{greedy}$: What type of trunk are you referring to? | The large storage box at the back of a car. 
 The large storage compartment of a car. 
 The back of a car. | 60% |
| | A large suitcase or box for storage. 
 The large, wooden storage chest. | 40% |
| $x_{QA}$: How many Grammy Awards does Whitney Houston have? 

 $q_{greedy}$: Are you referring to the number of Grammy Awards Whitney Houston won, or the number of Grammy Awards Whitney Houston was nominated for? | The number of Grammy Awards Whitney Houston won. *(Repeated × 4)* | 80% |
| | The number of Grammy Awards Whitney Houston was nominated for. | 20% |

# 6 EXPERIMENTS: DETERMINING WHEN TO CLARIFY

For our experiments on determining when to clarify, we use the same base LLMs as above for answering questions with an without clarification. We adapt existing methods for uncertainty estimation and chain-of-thought reasoning as baselines for this task. We begin this section by describing our novel approach to this subtask, before introducing baselines below.

## 6.1 INTENT-SIM

Unsupervised methods for uncertainty quantification in LLMs generally rely on estimating entropy over the output distribution, using high entropy to identify erroneous outputs (Kadavath et al., 2022; Kuhn et al., 2023). While these methods perform well at identifying incorrect predictions, they fail to identify *why* predictions are incorrect. Determining when to ask for clarification requires moving beyond simply identifying incorrect outputs and requires systems to attribute when uncertainty is the result of ambiguity. In our proposed method, INTENT-SIM, we disentangle these two factors by explicitly estimating the ambiguity of a given input, which we quantify as the entropy over simulated user intents.

**Input:** LM $M$, NLI model $N$, User input $\mathbf{x}$, sampling temperature $T$, and simulation count $S$.
**Output:** Entropy over simulated intents, $u$.

1: $\mathbf{q} \leftarrow$ **GreedySample**$(M, [\mathbf{x}])$
2: **for** $i \in \{1, \ldots, S\}$ **do**
3:    $\mathbf{a_i} \leftarrow$ **TempSample**$(M, [\mathbf{x}; \mathbf{q}], T)$
4: $G \leftarrow \emptyset$
5: **for** $i \in \{1, \ldots, S-1\}$ **do**
6:    **for** $j \in \{i+1, \ldots, S\}$ **do**
7:       left $\leftarrow N([\mathbf{q}; \mathbf{a_i}], [\mathbf{q}; \mathbf{a_j}])$
8:       right $\leftarrow N([\mathbf{q}; \mathbf{a_j}], [\mathbf{q}; \mathbf{a_i}])$
9:       **if** left is entailment and right is entailment **then**
10:         $G \leftarrow G \cup \{< i, j >, < j, i >\}$
11: $C \leftarrow \emptyset$
12: **for** $i \in \{1, \ldots, S\}$ **do**
13:    **if** $\mathbf{a_i} \notin c \;\; \forall c \in C$ **then**
14:       $C \leftarrow C \cup$ **DFS**$(G, a_i)$
15: $\widehat{P}(c|\mathbf{x}) \leftarrow \frac{|c|}{S}, \;\; \forall c \in C$
16: $u \leftarrow$ **Entropy**$(\widehat{P}(\cdot|\mathbf{x}))$

Figure 2: INTENT-SIM algorithm. We first sample a clarifying question and multiple responses from the LLM. We then construct a equivalence graph of responses, $G$, using NLI to determine equivalence. Finally, we identify disjoint subgraphs of $G$ with depth-first-search, representing distinct intents, and estimate the entropy over intents.

Figure 2 illustrates our method. Using the same few-shot prompt structure as in our responsiveness task (exact prompt in Appendix B), we condition on the user's request to greedily generate a clarifying question. We then simulate different user intents by sampling multiple responses to the clarifying question (example generations in Table 3). Following Kuhn et al. (2023), we then cluster sets of semantically equivalent responses using a DeBERTa-large NLI model (He et al., 2021) finetuned on MNLI Williams et al. (2018). We say that two responses are equivalent if both clarifying QA pairs entail each other, then estimate the likelihood of each set as the proportion of samples in it. Finally, we compute our uncertainty estimate by computing the entropy of this distribution over semantically distinct answers. In our experiments we decode 10 user responses with temperature $T = 0.5$ for all systems, following follow prior work on estimating uncertainty in LLMs from samples (Kuhn et al., 2023; Cole et al., 2023). Additional implementation details are provided in Appendix B.

Table 4: Results for determining when to clarify. We report AUROC and system performance under different interaction budgets ($b$), evaluated using contrastive accuracy for MT, accuracy (answer recall) for QA, and classification accuracy for NLI. We also report the percent gain in performance relative to the total gain from asking for clarification on all examples.

| Task | Model | Method | AUROC | $b = 10\%$ | $b = 20\%$ | $b = 30\%$ |
|---|---|---|---|---|---|---|
| MT | GPT-3 | Random | 0.500 | 76.8 (10%) | 78.6 (20%) | 80.4 (30%) |
| | | Likelihood | **0.547** | 76.1 (6%) | 78.1 (17%) | 79.8 (27%) |
| | | Self-Ask | 0.371 | 77.3 (13%) | **79.5 (25%)** | 81.5 (37%) |
| | | User Sim | 0.447 | **79.0 (22%)** | 79.3 (24%) | **82.1 (40%)** |
| NLI | LLaMA-2 7B Chat | Random | 0.500 | 42.4 (10%) | 43.8 (20%) | 45.2 (30%) |
| | | Likelihood | 0.416 | 41.2 (1%) | 40.0 (-7%) | 39.4 (-11%) |
| | | Self-Ask | 0.477 | 41.6 (4%) | 41.9 (7%) | 42.5 (11%) |
| | | User Sim | **0.564** | **43.3 (17%)** | **45.7 (33%)** | **49.9 (62%)** |
| | LLaMA-2 13B Chat | Random | 0.500 | 30.1 (10%) | 32.2 (20%) | 34.4 (30%) |
| | | Likelihood | 0.526 | 31.0 (14%) | **33.0 (24%)** | 33.8 (27%) |
| | | Self-Ask | 0.462 | 28.2 (1%) | 30.6 (12%) | 34.0 (28%) |
| | | User Sim | **0.532** | **31.4 (16%)** | 32.8 (23%) | **35.6 (36%)** |
| QA | GPT-3 | Baseline | 0.500 | 55.2 (10%) | 55.7 (20%) | 56.1 (30%) |
| | | Likelihood | 0.590 | **55.4 (14%)** | 55.9 (25%) | **56.3 (35%)** |
| | | Self-Ask | 0.538 | 55.1 (6%) | 55.6 (18%) | 56.2 (32%) |
| | | User Sim | **0.598** | **55.4 (14%)** | **55.9 (26%)** | **56.3 (35%)** |
| | LLaMA-2 7B Chat | Random | 0.500 | 38.9 (10%) | 39.4 (20%) | 39.9 (30%) |
| | | Likelihood | 0.510 | 38.4 (-1%) | 39.1 (14%) | 39.7 (28%) |
| | | Self-Ask | 0.510 | 38.9 (10%) | 39.3 (17%) | 39.9 (32%) |
| | | User Sim | **0.512** | **39.2 (16%)** | **40.3 (39%)** | **40.5 (43%)** |
| | LLaMA-2 13B Chat | Random | 0.500 | 41.2 (10%) | 41.6 (20%) | 42.1 (30%) |
| | | Likelihood | **0.551** | 41.1 (8%) | 41.7 (21%) | 41.8 (24%) |
| | | Self-Ask | 0.546 | 41.0 (6%) | 41.6 (20%) | 42.1 (30%) |
| | | User Sim | 0.550 | **41.6 (18%)** | **41.8 (23%)** | **42.6 (39%)** |

## 6.2 BASELINES

**Random** We simply report the expected performance of randomly selecting examples to clarify.

**Likelihood** For this baseline, we first prompt the model to generate the answer without clarification using the same few-shot prompt as above. We then use the likelihood of the greedily generated output to determine when to clarify. This simple yet effective baseline is often used for uncertainty estimation, determining which model outputs are likely incorrect. In this work, we use low-certainty in the output as an indicator that clarification may improve the model's response.

**Self-Ask** Introduced by Press et al. (2022), this prompting method is designed elicit chain-of-thought reasoning from LLMs for compositional reasoning tasks such as multi-hop QA. In their method, LLMs decompose inputs into multiple sub-questions and answers, which are composed to get the final answer. Self-Ask revolves around an intermediate step, where models decide whether to continue generating more questions or to complete their final response. We adapt this technique for our task, where the focus is not on decomposing the input but on querying for outside context. We adjust our few-shot prompt from our responsiveness task above and prompt assistants after each input query with the question "Is a follow-up question needed here?" (exact prompt in Appendix B). We then use the likelihood of generating "No" to score whether that clarification is needed. We also include this step in our sampled few-shot exemplars, creating a 50-50 split between unambiguous inputs, where the system responds "No", and ambiguous inputs, where systems respond "Yes".

## 6.3 RESULTS

In Table 4, we report results using our various LLMs and methods for deciding when to clarify. Looking at system performance under different interaction budgets, we observe that that Likelihood and Self-Ask demonstrate mixed results, occasionally performing worse than random under some

budget settings. In contrast, simulating user interactions consistently outperforms all baselines, and is the only system that always outperformans the random baseline under all interaction budgets.

Despite strong performances, we observe that this strong performance does not always translate similar gains in our AUROC metric. We attribute this gap to the coarse-grained nature of estimating entropy from samples. With only 10 samples, many examples produce the same distribution over clarifying answers equivalency sets. This is particularly true for examples where the entropy over clarifying answers is low. As a result, under very large values for $b$, entropy over simulated user responses may under perform compared to these other baselines; however, these larger values of $b$ are also less practical, as we aim for systems that ask questions conservatively, and $< 50\%$ of examples in both datasets benefit from clarification.

## 7 RELATED WORK

**Clarifying Questions** Selecting clarification questions has been previously studied in task-specific settings. Rao & Daumé III (2018) explores re-ranking clarification questions in product FAQ's, and Shridhar et al. (2023) studies generating clarifying questions as a supervised learning task, using generated questions for multi-step reasoning in knowledge distillation. Pyatkin et al. (2022) uses reinforcement learning (RL) to guide their question generation model toward proposing questions that can have a large effect on its moral judgment of a situation. Prior work (Yu et al., 2019) studies balancing asking clarification questions and making the final classification prediction over multiturn interactions. Their clarifying questions only cover existing attributes, while ours are open ended.

**Uncertainty Estimation** Several existing works have studied methods for disentangling different sources of uncertainty. Kamath et al. (2020) studies predicting out-of-distribution test examples, a source of epistemic uncertainty, and performing selective prediction by abstaining from predicting on such inputs. Kuhn et al. (2023) attempts to merge the likelihoods semantically equivalent in QA, eliminating the effect of uncertainty due to multiple vocalizations of the same answer. Cole et al. (2023) studies a similar setting in the intersection of ambiguity and uncertainty; however, this work does not consider the degree of ambiguity of various inputs, and does not attempt to resolve ambiguity through interaction. Other works have studied uncertainty estimation techniques for LLMs (Kadavath et al., 2022; Lin et al., 2022), but they do not explicitly model or evaluate their ability to disentangle different sources of uncertainty. These works also explore supervised methods for uncertainty estimation in LLMs, but find that these methods generalize poorly to new domains.

**Ambiguity in NLP** Numerous prior works have created datasets for studying ambiguity in NLP, including work in coreference resolution (Yuan et al., 2023), NLI (Pavlick & Kwiatkowski, 2019), and MT (Pilault et al., 2023). The last work on MT also studies resolving ambiguity in an interactive chain-of-thought setting; however, it does not consider the challenge of modeling how ambiguous a given input is or determining whether interaction is helpful. Ambiguity benchmarks can also provide a lens to study biases. Parrish et al. (2021) studies an ambiguous QA task where systems are evaluated whether they resolve ambiguity by relying on harmful social biases.

## 8 CONCLUSION

We present a unified framework for resolving ambiguity with clarifying questions, and apply it to QA, MT, and NLI. Our framework exposes the challenges in modeling clarifying interactions, and motivates the further study of disentangling uncertainty estimation and identifying when uncertainty can be attributed to ambiguity. We present a novel uncertainty estimation approach for this objective, INTENT-SIM, which we demonstrate improves detection of when to clarify.

Our framework lays the foundation for future work to explore interactive ambiguity resolution in general-purpose AI assistants. Future works using our framework may include developing new task-agnostic methods for generating clarification questions, or extending our framework to handle multi-turn interactions. Our work also motivates a closer examination of what is being learned through chat-finetuining for LLMs. Future work may develop new, multi-turn learning objectives using our framework and teach models to use interaction pragmatically, asking users clarifying question to maximize the accuracy of their responses.

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

## A    ADDITIONAL DATA DETAILS

In Table 5 include of raw examples from each of our MT, QA, and NLI datasets and in Table 6 we incude dataset statistics. In matching interpretations from NaturalQuestions to AmbigQA disambiguations, we eliminate all examples where the NaturalQuestions answers do not appear in any of the AmbigQA interpretations and when it matches more than one interpretation.

## B    ADDITIONAL MODELING DETAILS

### B.1    ORACLE QUESTION GENERATION SYSTEM

We present our oracle clarification generation prompts for QA, NLI, and MT in Table 7, Table 8, and Table 9, respectively. We do not provide GPT-3.5 any system prompt, and the entire body of the prompt is provided in a single user-side message. Note that for our MT oracle prompt, there is the risk of answer leakage, since the output translation is included in the prompt. However, we do not find this is an issue, as the generated followup questions and answers are always in the source language only.

### B.2    ADDITIONAL EXPERIMENTAL DETAILS AND RESULTS

**Prompts**    We present the prompts for responding with clarification, without clarification, and for SelfAsk in Tables 10, 12, and 11. These tables also demonstrate the variations in prompt between tasks, particularly in the instructions. We base our NLI instruction and class-to-token mapping on the prompts from Liu et al. (2023).

To perform our experiments with disambiguated inputs for QA and NLI, we use the same prompt as responding without clarificaiton, substituting the input with the disambiguate form of the input. For MT where disambiguations are given as additional context sentences, we simply prepend "Context: ..." onto each user input, filling in the context sentence.

For sampling unambiguous examples for SelfAsk, we use the unambiguous examples labeled in the MT and QA datasets. For NLI, where all examples are labeled as ambiguous, we use examples where all 9 annotators interpreted the input the same way as unambiguous examples, as these demonstrate the least variation in user intents.

When To Ask Results In Table 13, we present our full table of results on our determining when to ask for clarification subtask. Note that we only inlude instances where sytems improved by at least 10% with clarification, as noticeable gains are a prerequisite to this task.

Table 5: Raw ambiguous examples from each dataset.

| Task | Input ($x$) | Interpretations / Outputs ($y$) |
|---|---|---|
| MT | That is so sweet! | **Context:** You've been so wonderful to me these past couple of months.
**Target:** C'est tellement adorable. |
| | | **Context:** Try some - it's like a sugar explosion!
**Target:** C'est tellement sucré. |
| | I've never seen so much dough! | **Context:** The pizza's in the oven, but there's still some dough left.
**Target:** Je n'ai jamais vu autant de pâte ! |
| | | **Context:** Here you are - you've earnt it.
**Target:** Je n'ai jamais vu autant de thune ! |
| QA | When is episode 113 of dragon ball super coming out? | **Disambig:** When is episode 113 of dragon ball super coming out for its original airdate?
**Answer:** October 29, 2017 |
| | | **Disambig:** When is episode 113 of dragon ball super coming out for its american airdate?
**Answer:** June 1, 2019 |
| | Who plays the science officer on star trek discovery? | **Disambig:** Who plays the science officer on star trek discovery who is a chief engineer?
**Answer:** Anthony Rapp |
| | | **Disambig:** Who plays the science officer on star trek discovery who is a Kelpien?
**Answer:** Doug Jones |
| | | **Disambig:** Who plays science officer Michael Burnham on Star Trek Discovery?
**Answer:** Sonequa Martin-Green |
| NLI | A large number of people were not willing to take the risk. / A small number of people were willing to take the risk. | **Disambig:** A large number of people, but not all people, were not willing to take the risk.
**Label:** entailment |
| | | **Disambig:** A large number of people, and possibly all people, were not willing to take the risk.
**Label:** neutral |
| | We have not been able to find any scientific evidence that extraterrestrial life exists. / There is no scientific evidence that extraterrestrial life exists. | **Disambig:** There is no scientific evidence to be found that extraterrestrial life exists.
**Label:** neutral |
| | | **Disambig:** There has been no scientific evidence collected that extraterrestrial life exists.
**Label:** entailment |

Table 6: Counts of ambiguous and unambiguous inputs for each task. We also include counts of the number of sampled interpretations used in our "determining when to clarify" evaluations and the total number of interpretations.

| Task | Ambiguous $x$ | Unambiguous $x$ | Sampled Interpretations $y_*$ | Total Interpretations $y_*$ |
|---|---|---|---|---|
| NLI | 504 | 0 | 504 | 1008 |
| QA | 652 | 830 | 1482 | 2781 |
| MT | 88 | 176 | 352 | 352 |

Table 7: QA Followup Generation Prompt.

Given the Ambiguous Question and several possible Intended Interpretations, ask a Clarification Question and provide Clarification Responses corresponding to each Intended Interpretations. Here are two examples:

Example 1:
Ambiguous Question: Who has the highest goals in world football?
Intended Interpretation 1: Who has the highest goals in men's world international football?
Intended Interpretation 2: Who has the highest goals all-time in men's football?
Intended Interpretation 3: Who has the highest goals in women's world international football?

Clarification Question: Are you referring to the highest goals in men's world international football, or the highest goals in women's world international football?
Clarification Response 1: The highest goals in men's world international football.
Clarification Response 2: The highest goals all-time in men's football.
Clarification Response 3: The highest goals in women's world international football.

Example 2:
Ambiguous Question: Who won the last olympic men's hockey?
Intended Interpretation 1: Who won Olympic men's ice hockey in 2014?
Intended Interpretation 2: Who won Olympic men's ice hockey in 2010?
Intended Interpretation 3: Who won Olympic men's ice hockey in 2006?
Intended Interpretation 4: Who won the 2016 olympic men's field hockey?
Intended Interpretation 5: Who won the 2012 olympic men's field hockey?
Intended Interpretation 6: Who won the 2008 olympic men's field hockey?
Clarification Question: Which year? Are referring to field hockey or ice hockey?
Clarification Response 1: 2014, ice hockey.
Clarification Response 2: 2010, ice hockey.
Clarification Response 3: 2006, ice hockey.
Clarification Response 4: 2016, field hockey.
Clarification Response 5: 2012, field hockey.
Clarification Response 6: 2008, field hockey.

Now do it yourself:
Ambiguous Question: {}
Intended Interpretation 1: {}
. . .
Intended Interpretation $k$: {}

Table 8: NLI Followup Generation Prompt.

---

Given the Ambiguous Phrase and two possible Intended Interpretations, ask a Clarification Question and provide two Clarification Responses corresponding to each Intended Interpretations. Here are two examples:

Example 1:
Ambiguous Phrase: Jon will wash his car, and Mary will too.
Intended Interpretation 1: Jon will wash his car, and Mary will wash hers.
Intended Interpretation 2: Jon and Mary will both wash Jon's car.
Clarification Question: Will Jon and Mary wash the same or different cars?
Clarification Response 1: The same.
Clarification Response 2: Different.

Example 2:
Ambiguous Phrase: The hospital is being sued by six foot doctors.
Intended Interpretation 1: The hospital is being sued by six podiatrists.
Intended Interpretation 2: The hospital is being sued by doctors who are six feet tall.
Clarification Question: Do you mean six podiatrists or doctors who are six feet tall.
Clarification Response 1: Podiatrists.
Clarification Response 2: Doctors who are six feet tall.

Now do it yourself:
Ambiguous Phrase: {}
Intended Interpretation 1: {}
Intended Interpretation 2: {}

---

Table 9: MT Followup Generation Prompt.

---

Given the Ambiguous Phrase and two possible Translations, ask a Clarification Question about the meaning of a specific word or phrase and provide two Clarification Responses corresponding to each Translation. Here are two examples:

Example 1:
Ambiguous Phrase: Where are the bats?
Translation 1: Où sont les chauves-souris?
Translation 2: Où sont les battes?
Clarification Question: What type of bats do you mean?
Clarification Response 1: The small animals with wings.
Clarification Response 2: Sticks like you would use for a sport.

Example 2:
Ambiguous Phrase: I love dates.
Translation 1: J'adore les rendez-vous.
Translation 2: J'adore les dattes.
Clarification Question: What type of dates are you referring to?
Clarification Response 1: Meetings or appointments with someone.
Clarification Response 2: The sweet fruit from the date palm tree.

Now do it yourself:
Ambiguous Phrase: {}
Translation 1: {}
Translation 2: {}

---

Table 10: Prompt for few-shot MT with clarification questions. The colored text represents the output generated from the LLM assistant.

| Role | Messages |
|------|----------|
| System | Translate the English sentences into French. |
| User | English: It's tight enough if you ask me. |
| Assistant | Follow-Up Question: Are you referring to physical tightness or tension in a situation? |
| User | Follow-Up Response: I'm referring to a tense or stressful situation. |
| Assistant | French: C'est déjà assez tendu à mon avis. |
| … | |
| User | English: Why don't you care for them then? |
| Assistant | Follow-Up Question: What do you mean by 'care for them'? |
| User | Follow-Up Response: I mean, why don't you take responsibility for them or look after them? |
| Assistant | French: Alors pourquoi tu ne t'en occupes pas ? |

Table 11: Prompt for few-shot QA with SelfAsk. The colored text represents the output that is scored by LLM assistant, and used as an uncertainty estimate. Sampled exemplars are always half unambiguous examples, not requiring follow-up questions, and half ambiguous examples, requiring follow-ups.

| Role | Messages |
|------|----------|
| System | Answer the question. |
| User | Question: How many medals did australia win in the 2000 olympics? |
| Assistant | Is a Follow-Up Question Needed Here? Yes. Follow-Up Question: Are you asking about the number of gold, silver, bronze, or total medals that Australia won in the 2000 Olympics? |
| User | Follow-Up Answer: The total number of medals that Australia won in the 2000 Olympics. |
| Assistant | Answer: 58. |
| User | Question: Who plays erica on the last man on earth? |
| Assistant | Is a Follow-Up Question Needed Here? No. Answer: Cleopatra Coleman. |
| … | |
| User | Question: How long do contestants get to answer on jeopardy? |
| Assistant | Is a Follow-Up Question Needed Here? No. |

Table 12: Few-shot NLI prompt without clarification. The colored text represents the output generated from the LLM assistant.

| Role | Messages |
|------|----------|
| System | For each Context, determine whether the Claim is True, False, or Inconclusive. |
| User | Context: The professor admitted that no students wrote course evaluations, which is surprising.
Claim: It is surprising that no students wrote course evaluations. |
| Assistant | Answer: True. |
| ... | |
| User | Context: Many birds are attracted to the island because of the abundance of food.
Claim: Many birds are attracted to the island because of the abundance of trees. |
| Assistant | Answer: Inconclusive. |

Table 13: Full results on our determining when to ask for clarification subtask. Above, we report AUROC as well as system performance (answer recall for QA and contrastive accuracy for MT) under different interaction budgets ($b$), allowing systems to ask for clarification on only the top $b\%$ of examples. We also report the percent gain in performance relative to the total gain from asking for clarification on all examples. Here, we limit results to systems that demonstrate strong responsiveness to clarification, as being able to utilize clarifying questions is a prerequisite for determining when clarification is useful.

| Task | Model | Method | AUROC | $b = 10\%$ | $b = 20\%$ | $b = 30\%$ |
|------|-------|--------|-------|-----------|-----------|-----------|
| MT | GPT-3 | Baseline | 0.500 | 76.8 (10%) | 78.6 (20%) | 80.4 (30%) |
| | | Likelihood | **0.547** | 76.1 (6%) | 78.1 (17%) | 79.8 (27%) |
| | | Self-Ask | 0.371 | 77.3 (13%) | **79.5 (25%)** | 81.5 (37%) |
| | | User Sim | 0.447 | **79.0 (22%)** | 79.3 (24%) | **82.1 (40%)** |
| NLI | LLaMA-2 7B Chat | Baseline | 0.500 | 42.4 (10%) | 43.8 (20%) | 45.2 (30%) |
| | | Likelihood | 0.416 | 41.2 (1%) | 40.0 (-7%) | 39.4 (-11%) |
| | | Self-Ask | 0.477 | 41.6 (4%) | 41.9 (7%) | 42.5 (11%) |
| | | User Sim | **0.564** | **43.3 (17%)** | **45.7 (33%)** | **49.9 (62%)** |
| | LLaMA-2 13B Chat | Baseline | 0.500 | 30.1 (10%) | 32.2 (20%) | 34.4 (30%) |
| | | Likelihood | 0.526 | 31.0 (14%) | **33.0 (24%)** | 33.8 (27%) |
| | | Self-Ask | 0.462 | 28.2 (1%) | 30.6 (12%) | 34.0 (28%) |
| | | User Sim | **0.532** | **31.4 (16%)** | 32.8 (23%) | **35.6 (36%)** |
| QA | GPT-3 | Baseline | 0.500 | 55.2 (10%) | 55.7 (20%) | 56.1 (30%) |
| | | Likelihood | 0.590 | **55.4 (14%)** | 55.9 (25%) | **56.3 (35%)** |
| | | Self-Ask | 0.538 | 55.1 (6%) | 55.6 (18%) | 56.2 (32%) |
| | | User Sim | **0.598** | **55.4 (14%)** | **55.9 (26%)** | **56.3 (35%)** |
| | LLaMA-2 7B | Baseline | 0.500 | 34.9 (10) | 35.5 (20) | 36.1 (30) |
| | | Likelihood | **0.533** | **35.0 (11%)** | **35.4 (19%)** | **36.4 (34%)** |
| | | Self-Ask | 0.486 | 34.5 (4%) | 35.3 (16%) | 35.9 (26%) |
| | | User Sim | **0.533** | **35.0 (11%)** | **35.4 (19%)** | **36.4 (34%)** |
| | LLaMA-2 7B Chat | Baseline | 0.500 | 38.9 (10%) | 39.4 (20%) | 39.9 (30%) |
| | | Likelihood | 0.510 | 38.4 (-1%) | 39.1 (14%) | 39.7 (28%) |
| | | Self-Ask | 0.510 | 38.9 (10%) | 39.3 (17%) | 39.9 (32%) |
| | | User Sim | **0.512** | **39.2 (16%)** | **40.3 (39%)** | **40.5 (43%)** |
| | LLaMA-2 13B | Baseline | 0.500 | 42.7 (10%) | 43.4 (20%) | 44.0 (30%) |
| | | Likelihood | 0.593 | **43.0 (15%)** | 43.7 (24%) | 44.6 (39%) |
| | | Self-Ask | 0.504 | **43.0 (15%)** | 43.5 (21%) | 44.0 (29%) |
| | | User Sim | **0.671** | **43.0 (15%)** | **44.5 (38%)** | **45.9 (59%)** |
| | LLaMA-2 13B Chat | Baseline | 0.500 | 41.2 (10%) | 41.6 (20%) | 42.1 (30%) |
| | | Likelihood | **0.551** | 41.1 (8%) | 41.7 (21%) | 41.8 (24%) |
| | | Self-Ask | 0.546 | 41.0 (6%) | 41.6 (20%) | 42.1 (30%) |
| | | User Sim | 0.550 | **41.6 (18%)** | **41.8 (23%)** | **42.6 (39%)** |

