# OpenReview forum: "Clarify When Necessary: Resolving Ambiguity with Language Models"
_ICLR.cc/2024/Conference — ICLR 2024 Conference Withdrawn Submission_

### Official Review · Reviewer_R4of · 2023-10-26

**Soundness:** 2 fair
**Presentation:** 3 good
**Contribution:** 2 fair
**Rating:** 3
**Confidence:** 3

**Summary:**

The paper investigates the task of predicting whether it would be useful for an LLM to ask a clarifying question in order to answer to a particular input. The method samples different outputs from the model, clusters them using a separate entailment model, and then uses the cluster sizes to assign uncertainty and decide about asking a question. Evaluation is performed on three datasets (QA, MT, NLI) that contain manually created alternative disambiguations for each ambiguous input.

**Strengths:**

The topic is interesting.
The idea of assessing the need for a clarification question based on downstream output distribution seems quite good.
Evaluation on 3 different tasks and datasets is nice to see.

**Weaknesses:**

Based on my understanding of the paper, there is a serious methodological issue in the setup. In order to measure the uncertainty and decide whether a clarifying question is needed, the setup seems to use the manually created disambiguated intents from the datasets. Section 5.1, paragraph 2 refers to Appendix B for the full prompt and those prompts use the disambugations as input.

In this example from the appendix, the generated clarification question and the implicit ambiguity that it tries to resolve are clearly guided by the provided intended interpretations, which have been manually created for that particular dataset:

Ambiguous Phrase: Jon will wash his car, and Mary will too.
Intended Interpretation 1: Jon will wash his car, and Mary will wash hers.
Intended Interpretation 2: Jon and Mary will both wash Jon’s car.
Clarification Question: Will Jon and Mary wash the same or different cars?

Given that, this proposed method could not be used in a realistic setting, as it relies on a manually created set of disambiguations for a given input, which would not be available for any new input outside of the described datasets.

Furthermore, the manually created intents in the prompt leak information into the generated questions and answers. The evaluation therefore measures whether it would be good for an unrealistic oracle to ask a question, not whether it would be good for a realistic system to ask a question.


Evaluating MT using only contrastive accuracy is problematic. The model can be quite far from the correct answer and still be counted as correct using this method. Some more common MT evaluation metric would have been reported as well.

The paper presents the separation of the task into 3 different subtasks as a contribution, although this has been done in previous works as well (https://arxiv.org/abs/2305.15933).

The paper has quite a large number of writing issues and grammatical errors, mainly preposition errors or missing/repeated words. The overall clarity of the paper and the technical details could be improved.

**Questions:**

Section 2 details that only aleatoric uncertainty should be measured for clarification. Why do you feel that epistemic uncertainty (missing knowledge) could not also be helped with clarifying questions?

Why are NLI results in the disambig setting identical regardless of whether they are uniform or sampled?

Table 4 refers to user sim, the paper text refers to intent-sim. I suspect that is not intentional.

---

> ### Author Response · Authors · 2023-11-22
>
> Thank you for your response!
>
> In our work, we are clear with distinguishing our system for generating clarifying questions as an oracle-type system that uses known disambiguations and is, therefore,  not always practical; however, this is not the focus of our work. While generating clarifying questions has been previously explored in existing work, methods and evaluations for the other two subtasks (responding to clarification and determining when to ask for clarification) have not received as much attention. By using an oracle to generating high-quality clarifying questions and their respective answers, we are able to establish evaluations for evaluating these other two subtasks. Furthermore, we are able to avoid any answer leakage that could be involved in generating clarifying questions or responses by not providing output to systems when generating clarifying questions or answers. The only exception to this is with our MT prompt, however, we observe that no clarifying questions nor answers include words from the target language and therefore do not suffer from leakage.
>
> We agree that including a discussion on standard MT evaluation metrics, in addition to BLEU score, would be beneficial for understanding true model behaviors in these settings, and plan to include these in our revisions (GPT3 gets and average BLEU of 0.318 with clarifications and 0.412 without clarifications). Given the relatively small number of test examples and challenges in evaluating these subtle translation phenomena using standard metrics (see “Does Neural Machine Translation Benefit from Larger Context?”, Jean et al., 2017 for further discussion), we rely on a more targeted evaluation.

---

### Official Review · Reviewer_Mm3G · 2023-10-29

**Soundness:** 3 good
**Presentation:** 4 excellent
**Contribution:** 2 fair
**Rating:** 5
**Confidence:** 3

**Summary:**

This paper proposes a new testbed for evaluating a system's ability to ask clarification questions, where a system needs to decide 1) when to ask a clarification question, and 2) ask a clarification question (though not evaluated in this paper), and 3) given a clarification question and a response, provide an answer. The paper re-purposed existing datasets and collected additional annotation in three tasks: QA, NLI, and MT, and designed automatic evaluation metrics for these tasks. The paper additionally proposed a new method, Intent-Sim, which scores the uncertainty by estimating the answer entropy of a greedily decoded question.

**Strengths:**

The paper is generally clearly written. The tasks and datasets are well-motivated, and the evaluation results are comprehensive.

**Weaknesses:**

1. It'd be helpful to know how well humans perform on this task (e.g., AUC ROC for deciding whether to ask clarification questions).
2. It'd be also useful to evaluate GPT-4 on this task. This task feels straightforward to me and I suspect that GPT-4 is on par with human-level (I could be wrong though)
3. I am uncertain whether Intent-Sim actually helps -- In table 2, it seems that it does not significantly outperform the random baseline?
4. The paper did not evaluate the quality of the clarification question. While I do agree with the authors that they are hard to evaluate and control, I feel that this is the most challenging and interesting part of the task.

**Questions:**

^ See weaknesses 1-3 above.

---

> ### Author Response · Authors · 2023-11-22
>
> Thank you for your review!
>
> With respect to your comments regarding establishing human performance, we agree that determining whether humans can determine when a LLM should ask for clarification is an interesting line of inquiry; however, it does not serve as an upper-bound/target benchmark for systems as human baselines often are.
>
> Regarding using GPT-4 as an additional baseline, we agree that understanding this system’s performance would be interesting; however, LLAMA-2 and GPT-3 performance together paint a clear picture of the current state of systems at these tasks. Furthermore, output likelihoods are not available from GPT-4 and ChatGPT systems, making some baselines for these systems impossible to evaluate. While improvements under different interaction budgets appear small, this is, in part, due to the nature of these evaluations where gains are limited by both the budget and system performance with clarification. Increasing (1) the proportion of ambiguous to unambiguous inputs and (2) the ability of LLMs to determine the correct output with clarification would increase these margins.
>
> Finally, we agree that generating useful clarifying questions is a very interesting and important line of research that we do not focus on in this work. Prior work has explored the task of generating clarifying questions; however, relatively little attention has been paid to these additional tasks of (1) responding to clarifications and (2) determining when clarification questions are necessary. The focus of this work is to provide the framework for a comprehensive solution for asking clarifying questions, and focusing providing evaluation benchmarks for these other two tasks.

---

### Official Review · Reviewer_AZhU · 2023-10-31

**Soundness:** 2 fair
**Presentation:** 2 fair
**Contribution:** 2 fair
**Rating:** 3
**Confidence:** 4

**Summary:**

This paper proposes a framework for resolving ambiguity which consists of three subtasks.  The first task is to decide when the clarification question is needed. The second task is to determine what clarification question to task. And the third task is to generate the final response conditioning on the input and the clarifying QA pair. The authors apply their framework on QA, MT and NLI tasks. The experimental results show that with clarifying QA pairs and disambiguated inputs, the downstream task performance tends to increase. Furthermore,  INTENT-SIM is proposed which explicitly estimates the ambiguity by calculating the entropy over user intents. Results show that the proposed system always outperforms the random baseline under all interaction budgets.

**Strengths:**

- Some common ambiguity types are summarized and analyzed, which could inspire future search.
- Multiple NLP tasks are evaluated.
- Multiple data generation settings are evaluated.

**Weaknesses:**

- Clarification question generation is based on the oracle prompting method, which may not be feasible for real-world applications.
- The compared baselines seem to be weak. The authors cite Cole et al. 2023 in Section 2 but do not compare with their method. Additionally, the improvements presented in Table 4, particularly regarding User similarity over Random, sometimes appear to be statistically insignificant.
- I understand that the authors try to show the generalization of the proposed framework on multiple NLP domains. But the setup for NLI and MT seems to be unnatural. It also makes the comparison across different tasks difficult. It is more intuitive to experiment on more QA datasets or dialogue systems, like [1].

[1] Prompting and Evaluating Large Language Models for Proactive Dialogues: Clarification, Target-guided, and Non-collaboration, Findings of EMNLP 2023

**Questions:**

- Why did you choose to conduct experiments on NLI and MT tasks rather than on more QA datasets or dialogue systems, where clarification questions naturally occur?
- Why not compare with the method proposed in Cole et al. 2023?

---

> ### Author Response · Authors · 2023-11-22
>
> Thank you for your thoughtful review!
>
> Re: Oracle clarification question generation setting
> We agree that, in practical settings, having generating clarifying questions based on known disambiguations is not always feasible; however, this is not the focus of our work (Section 2.2). While generating clarifying questions has been previously explored in existing work (Pyatkin et al., 2022; Rao & Daumé III, 2018), methods and evaluations for the other two subtasks (responding to clarification and determining when to ask for clarification) has not been studied in the open-ended settings explored in this work.
>
>
> Re: Choice of tasks / benchmarks
> While follow-up questions are prevalent in existing dialogue datasets, many questions are used for generating engaging conversations or determining a fixed set of parameters to fulfill a request (e.g., flight destinations, times). Most standard QA datasets, for ease of evaluation, select unambiguous questions which do not need clarifying information. Thus, we focus on the AmbigQA dataset which focuses on ambiguous questions. We chose additional benchmarks (NLI, MT) to study open-ended ambiguities. To this end, we investigate tasks such as NLI and MT for asking clarification questions since these are tasks where (1) ambiguities arise and (2) LLMs are frequently asked to perform.
>
> Re: Choice of baseline
> Regarding your concerns over system performance and our choices in baselines, we agree that comparing against other baselines would be beneficial. We look at the semantic entropy method from Kuhn et al., 2023, which builds upon the work method introduced by Cole et al., 2023 and computes the entropy over equivalent, sampled responses from the LLM and will include these results in our revisions.
>
> We would also like to emphasize that our results are primarily meant to illustrate the difficulty in this task of disentangling different sources of uncertainty. While improvements over the random baseline under different interaction budgets appear small, this is, in part, due to the nature of these evaluations where gains are limited by both the budget and system performance with clarification. Increasing (1) the proportion of ambiguous to unambiguous inputs and (2) the ability of LLMs to determine the correct output with clarification would increase these margins.

---

> > ### Comment · Reviewer_AZhU · 2023-11-22
> >
> > Thanks for the authors' response! Regarding the choice of tasks, I am still not convinced about the motivation for experimenting on additional benchmarks like NLI and MT to evaluate open-ended ambiguities. This setting seems too broad, as any NLP problem could involve ambiguity (e.g., NER, relation classification, ...). Each task has its unique aspects, and proposing a method that fits different settings could be challenging. I would suggest the authors focus on only one specific task type (e.g., QA).

---

### Official Review · Reviewer_SMVw · 2023-11-01

**Soundness:** 3 good
**Presentation:** 3 good
**Contribution:** 3 good
**Rating:** 8
**Confidence:** 4

**Summary:**

This paper focuses on the problem of handling ambiguous inputs in large language models. Specifically, it addresses three sub-problems: determining when to clarify model inputs, deciding on what clarifying questions to ask, and how to incorporate information obtained from clarification. The paper introduces a framework that analyzes common types of ambiguities across three different tasks and how large language models can use clarifying interactions. And, the paper finds that clarifying interactions can effectively enhance model performance and introduces a new method, INTENT-SIM, to measure model uncertainty and thereby decide when to pose clarifying questions. Experimental results indicate that the proposed method consistently outperforms the random baseline across different interaction budgets.

**Strengths:**

1. The research presented in this paper is well-motivated. The authors provide a clear description of the research problem, and the entire paper is well-organized and easy to follow.
2. The ambiguity analysis framework proposed by the authors is both simple and easily implementable. Furthermore, they demonstrate its effectiveness across various datasets and tasks, showing that the use of clarifying interactions indeed improves model performance.
3. I also think that the INTENT-SIM algorithm is innovative, and the strong empirical results in the experiments support its effectiveness.

**Weaknesses:**

I have concerns about the complexity of the INTENT-SIM algorithm. As it involves constructing graphs using external models and graph computations, this could significantly impact the computational efficiency of large language models.

**Questions:**

Why were these three specific tasks chosen for testing?

---

> ### Author Response · Authors · 2023-11-22
>
> Thank you for your thoughtful comments!
>
> Regarding your concern about the computational cost of running an external model and graph clustering in our Intent-Sim algorithm: while this does add additional computational overhead of computing confidence, in practice the inference cost is still dominated by the cost of inference with the LLM. Sampling multiple outputs from the LLM further increases these costs, however, and we plan to include a discussion of this in our revisions.

---

> > ### Comment · Reviewer_SMVw · 2023-11-23
> >
> > I have read the author's response and will keep my scores unchanged.